ecology/theoretical biology/health and disease and epidemiology

corona, epidemics, transmission, ecological dynamics, Leslie theory, epidemiology

**Authors for correspondence:**
Steinar Engen
e-mail: steinar.engen@ntnu.no
Nils Chr. Stenseth
e-mail: n.c.stenseth@ibv.uio.no

# The ecological dynamics of the coronavirus epidemics during transmission from outside sources when $R_0$ is successfully managed below one

Steinar Engen[1], Huaiyu Tian[2], Ruifu Yang[3], Ottar N. Bjørnstad[4], Jason D. Whittington[5] and Nils Chr. Stenseth[5]

[1]Centre for Biodiversity Dynamics (CBD), Department of Mathematical Sciences, Norwegian University of Science and Technology, 7491 Trondheim, Norway
[2]State Key Laboratory of Remote Sensing Science, College of Global Change and Earth System Science, Beijing Normal University, Beijing 100875, People's Republic of China
[3]State Key Laboratory of Pathogen and Biosecurity, Beijing Institute of Microbiology and Epidemiology, Beijing 100071, People's Republic of China
[4]Center for Infectious Disease Dynamics, Pennsylvania State University, University Park, PA 16802, USA
[5]Center for Ecological and Evolutionary Synthesis (CEES), Department of Biosciences and Faculty of Mathematics and Natural Sciences, University of Oslo, PO Box 1032 Blindern, 316 Oslo, Norway

SE, 0000-0001-5661-1925; HT, 0000-0002-4466-0858; ONB, 0000-0002-1158-3753; JDW, 0000-0002-4070-8658; NCS, 0000-0002-1591-5399

Since COVID-19 spread globally in early 2020 and was declared a pandemic by the World Health Organization (WHO) in March, many countries are managing the local epidemics effectively through intervention measures that limit transmission. The challenges of immigration of new infections into regions and asymptomatic infections remain. Standard deterministic compartmental models are inappropriate for sub- or peri-critical epidemics (reproductive number close to or less than one), so individual-based models are often used by simulating transmission from an infected person to others. However, to be realistic, these models require a large number of parameters, each with its own set of uncertainties and lack of analytic tractability. Here, we apply stochastic age-structured Leslie theory with a long history in ecological research to provide some new insights to epidemic dynamics fuelled by external imports. We model the dynamics of an epidemic when $R_0$ is below one, representing COVID-19 transmission following the

successful application of intervention measures, and the transmission dynamics expected when infections migrate into a region. The model framework allows more rapid prediction of the shape and size of an epidemic to improve scaling of the response. During an epidemic when the numbers of infected individuals are rapidly changing, this will help clarify the situation of the pandemic and guide faster and more effective intervention.

# 1. Introduction

The ongoing COVID-19 pandemic, caused by the SARS-CoV-2 virus, was first reported in Wuhan in China at the end of 2019, and has now spread throughout the world [1,2]. Genomic and phylogenetic analyses have shown that the virus has a zoonotic origin with a structure similar to several bat-derived coronaviruses [3–5]. China and the rest of the world have been implementing various non-pharmaceutical interventions in the months since first emergence [6,7]. When strictly applied, the implementation of these control measures successfully pushed the epidemic from a super-critical ($R_0 > 1$) to a subcritical ($R_0 < 1$) regime in several countries. In Wuhan, after the epidemic peaked in the first three months of 2020, most new cases were imports from abroad and further transmission was greatly suppressed using a mixture of testing, contact tracing and quarantine procedures. The effect of the interventions in China has shown that timely diagnosis and non-pharmaceutical protective countermeasures can significantly diminish and contain the spread of this virus.

In addition to monitoring the impact of intervention measures and studying the impact of asymptomatic carriage, we need to improve our understanding of plausible scenarios for what COVID-19 dynamics will look like in a post-pandemic era [8]. As local infection rates have declined in some areas due to the success of social-distancing efforts, and as vaccines reach an increasing proportion of populations, discussion over the removal or reduction in severity of interventions has grown [9], and it becomes important to evaluate the transmission dynamics that we may see after a reduction in $R_0$ values below those seen early in the pandemic.

In this paper, we present analyses aiming at improving our understanding of the dynamics of the epidemics when $R_0$ is below one, applicable to (i) a setting of migration into a region that has attained successful control (country of region within a country) from outside the region. In addition, our analysis will be relevant to (ii) the values of $R_0$ that countries are aiming for through successful application of interventions and (iii) the transmission from animal reservoirs to the human population which also represent migration from an outside source. While our approach is theoretical, it can clarify what the epidemic might look like when we look at data on immigration of new COVID-19 infections into countries to show what the sub- or peri-critical epidemic dynamics may look like for a range of $R_0$ values.

In this contribution, we apply a rich probabilistic framework to analyse effects of parameter and process uncertainty in a transparent way using an approach largely adopted in ecological studies. We use an age-structured model, where age now is interpreted as time since infection, which benefits from a long tradition of analysis of such stochastic models in ecology and mathematical demography. The theory goes back to the classical deterministic models of Fisher [10] and Leslie [11]. An age-structured population model is well understood theoretically, can accommodate the different uncertainties and can be rapidly evaluated, which is especially important when local health systems and resources are being stressed, such as during the ongoing COVID-19 pandemic.

To make ecological models realistic, it is usually necessary to introduce environmental noise, meaning that the Leslie matrices themselves are stochastic. Important theoretical progress was done by Tuljapurkar [12] introducing the concept of environmental variance in such models, and its effect on the long-term growth rate. Using diffusion approximations, the time to extinction in such models appears to follow approximately an inverse Gaussian distribution [13,14]. Later demographic stochasticity has also been analysed in such models using the concept of demographic variance in age-structured dynamics [15,16]. One important lesson to learn from these studies is that, even if there are a large number of parameters required to describe the means, variances and covariances in the Leslie matrix, the dynamics can be described very accurately by only three key parameters, the growth rate $\lambda$ and the environmental and demographic variances. In the present discussion we focus on demographic stochasticity [14], leaving us with only two parameters determining the distribution of time to extinction in a model without importation, and three parameters for the endemic stationary distribution in the presence of external seeding.

## 2. Model

We analyse the stochastic dynamics of the number of transmitters of the virus using an age-structured population model. For our analysis, we assume that dynamics happen in discrete time-steps, which is, for example, assumed in the chain-binomial epidemic model [17]. Leslie matrix theory, deterministic as well as stochastic, is a well-established field in ecological studies of age-structured populations. Here we apply this theory to study the dynamics of numbers of transmitters of an infectious disease like the COVID-19 epidemics. We do this by replacing the age of a person by the time since infection measured in days, birth rates at different ages by the rates of infection (the average number a transmitter infects during a day), and assume that the survivals, up to the 'age' at which the individual no longer transmits the disease transmissions, are 1. This last assumption is not crucial because the death rates during the infectious period are small, and infection rates are average numbers so that also dead people in general may contribute (with zero) to this while we consider them theoretically as being alive. In appendix A, we give the basic definitions and results for deterministic and stochastic Leslie matrix theory required for our analysis.

We write $L$ for the relevant $(k \times k)$ Leslie matrix for changes during a day, with daily transmission rates in the first row, survivals 1 at the sub-diagonal and other elements being zero, and let $n$ be the vector with elements equal to the number of infectious individuals in the different 'age' groups. The numbers are propagated forward in time through the matrix multiplication $Ln$. Then the number of infectious individuals grows approximately exponentially with multiplicative rate $\lambda$, which is the dominant real eigenvalue of $L$. After a transient initial period the number of infectious individuals will reach a stable 'age'-distribution $u$ with $\sum u_i = 1$, expressing the fractions of individuals in the different 'age' classes, which is the right eigenvector associated with $\lambda$. Based on the results of Du *et al.* [18] we shall assume that the transmission rates in the first row are proportional to a gamma distribution with mean 6.6 and shape parameter 1.87. To obtain a model with a specific value of $R_0$ the rates must be scaled by a common factor so that they sum up to $R_0$. If $R_0 = 1$, then $\lambda = 1$ and the number of infectious individuals is constant in the absence of stochasticity. This number is increasing ($\lambda > 1$) if $R_0 > 1$ and decreasing ($\lambda < 1$) if $R_0 < 1$. Figure 1a shows the parameter $r = \lambda - 1$ as function of $R_0$ for this model.

The left eigenvector $v$, scaled so that $\sum u_i v_i = 1$, has components called the reproductive values of the 'age'-classes, introduced by Fisher [10]. The total reproductive value of the number of infectious individuals, $V = \sum v_i n_i$, is approximately equal to the actual total number, but it has the advantage of being much simpler to analyse mathematically. In particular, when the transmissions are stochastic, for example, if one infected person during a day transmits the disease to a Poisson distributed number of person, the process $V$ can be approximated by a diffusion process with infinitesimal mean and variance $rV$ and $\sigma_d^2 V$, respectively. Here $r = \lambda - 1$ and $\sigma_d^2$ is called the demographic variance for the process. This approximation can be used also if there is over-dispersion $D$ (ratio between variance and mean) in transmissions relative to the Poisson model. The demographic variance is then $\sigma_1^2 D$, where $\sigma_1^2$ is the demographic variance under the Poisson assumption. The Poisson assumption may be seen as a large-population approximation to the chain-binomial epidemic model [17] and is also the assumption employed in $\tau$-leap simulation of event-based epidemics [19]. Figure 1b shows $\sigma_1^2$ as function for $R_0$ under the gamma model we are using. For more details on the model and parameter definitions see Appendix A.

For a process with initially $N_0 \approx V_0$ number of infected at time zero, the probability that extinction at time $T$ occurs before time $t$ is [13]

$$P(T < t) = e^{-2N_0 r e^{rt} / [\sigma_d^2 (e^{rt} - 1)]}, \tag{2.1}$$

for $r \neq 0$ and $e^{-2N_0/(\sigma_d^2 t)}$ for $r = 0$. Stochastic simulations have demonstrated that this approximation is sufficiently accurate for realistic values of $\sigma_d^2$ [15]. This formula for the distribution of time to extinction can be used to perform a sensitivity analysis for any range of values of the parameters $R_0$, $N_0$ and $D$. Some illustrating examples are shown in figure 2 using reference values $R_0 = 0.9$, $N_0 = 2500$ and $D = 1$, varying a single parameter in each graph.

## 3. Immigration

The above model is a so-called 'closed epidemic' model because it assumes no susceptible recruitment and no migration. If a country opens up for immigration it is interesting to see the effects of an average immigration rate (also referred to as the importation rate in epidemiology) $\mu$ of infected per

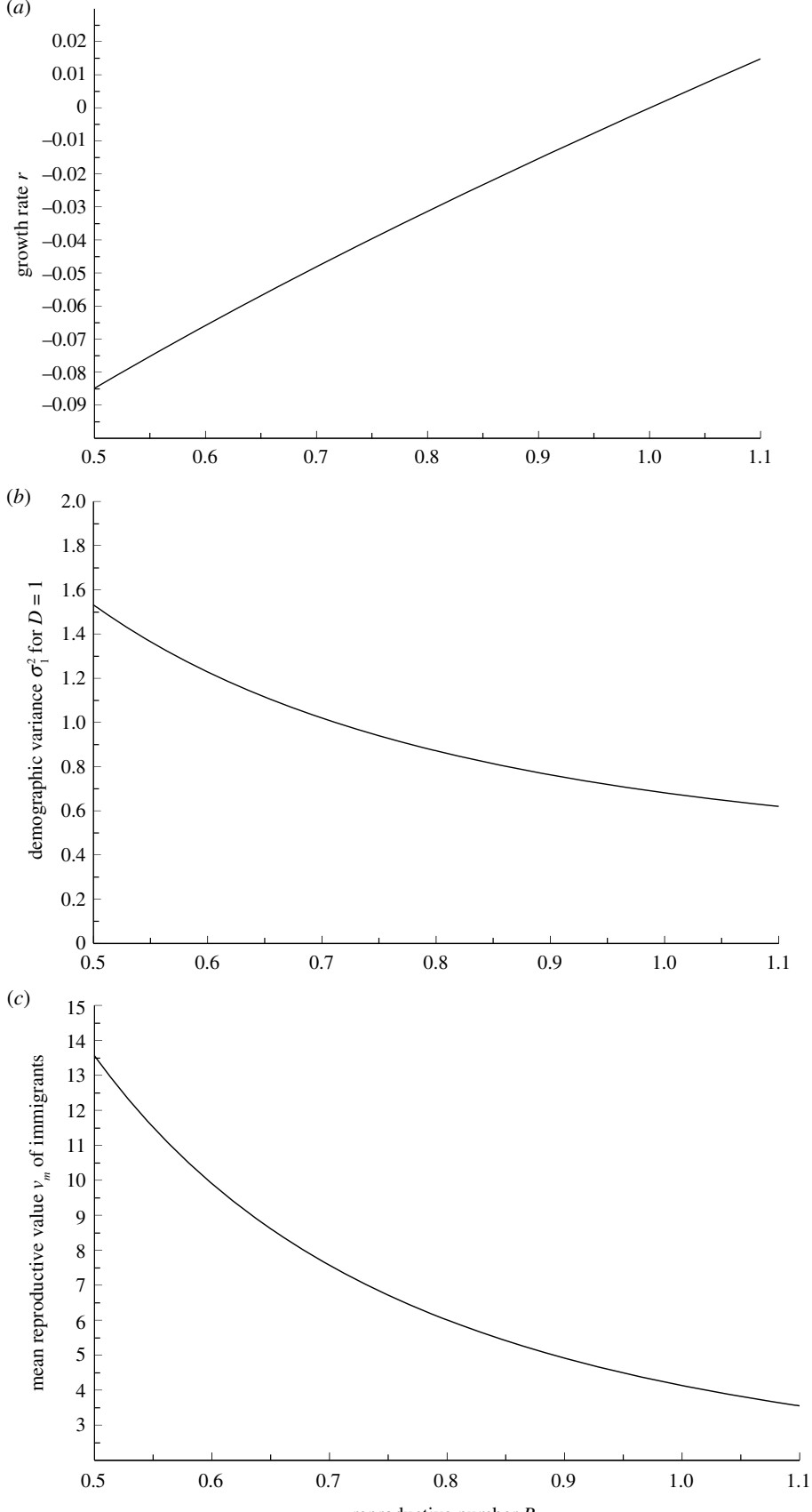

**Figure 1.** Panel (*a*) shows the growth rate *r* as function of $R_0$ using the gamma model for infection rates, with rates proportional to a gamma distribution with mean 6.6 and shape parameter 1.87. Panels (*b*, *c*) show $\sigma_1^2 = \sigma_d^2/D$ and mean reproductive value for immigrated cases assumed to be infected less than 4 days, $v_m = (v_1 + v_2 + v_3)/3$ for the same model.

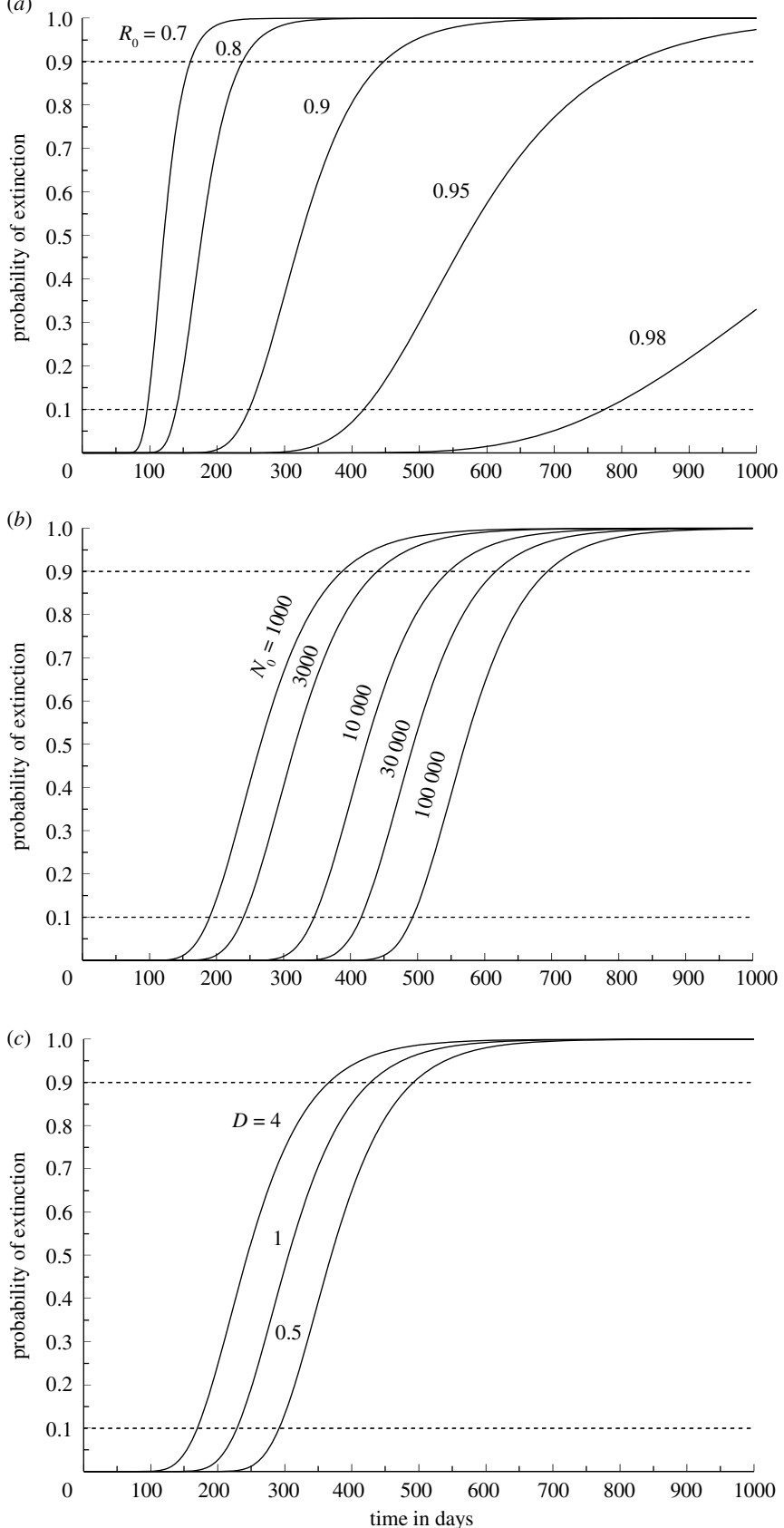

**Figure 2.** Probability of extinction of the epidemics under no imported infections as function of time in days. Standard set of parameters used are $N_0 = 2500$, $D = 1$, and $R_0 = 0.9$. One parameter is varied at the time, $R_0$ in (*a*), $N_0$ in (*b*), and $D$ in (*c*). The transmission rates are as in figure 1. The horizontal dashed lines at 0.9 and 0.1 can be used to read at which times the probability that the epidemics has gone extinct is 0.9 and 0.1, respectively.

day. With immigration the epidemic process will no longer go extinct even if $R_0 < 1$ but reach an equilibrium with stochastic fluctuation balancing the increase due to immigrations and the decrease due to $R_0 < 1$. To find the stationary distribution of $V$ describing these fluctuations we need to know the mean reproductive value $v_m$ of immigrants. In our illustrations, we choose this to be the mean reproductive value the first three days after infection. The infinitesimal mean of the total reproductive value process is then $rV + \mu v_m$. Notice that this represents the mean reproductive value of immigrants calculated from the Leslie matrix determining the transmission from infectious individuals. There will also be stochasticity in the immigration process. Accounting for over-dispersion $D_m$ relative to a Poisson distributed number of infected immigrants, the infinitesimal variance is $\sigma_d^2 V + v_m^2 D_m \mu$. Using the general formula for the stationary distribution in diffusion theory [13], this is

$$f(v) = C \, e^{2rv/\sigma_d^2} \left( 1 + \frac{\sigma_d^2 v}{v_m^2 D_m \mu} \right)^{2v_m \mu/\sigma_d^2 - 2rv_m^2 D_m \mu/\sigma_d^4 - 1}, \tag{3.1}$$

where $C$ is a constant chosen so that $f(v)$ integrates to one. The expected total reproductive value can be found directly from the infinitesimal mean that on average must be zero, giving $EV = -\mu v_m/r$. The variance in this distribution can be derived by writing a balance equation requiring that $V$ and $V + dV$ have the same variance [19]. This yields the stationary variance

$$\text{var}(V) = \frac{(-\sigma_d^2/r + v_m D_m)EV}{2} = \frac{(-\sigma_1^2 D/r + v_m D_m)EV}{2}. \tag{3.2}$$

Hence, $\text{var}(V)/EV$ is independent of the immigration rate $\mu$. In appendix B, we also derive all higher order cumulants for the stationary distribution confirming the result for the variance found by the method described above.

In addition to the properties of the stationary distribution, the process is also characterized by the speed of the fluctuations around the equilibrium. In deterministic theory this is often expressed by the characteristic return time to equilibrium defined as the time it takes for a perturbation away from the equilibrium to be reduced to a fraction $^1/e \approx 0.37$ of its original value. In stochastic theory the speed of fluctuations can be described by the temporal autocorrelation function $\rho(h)$ between the states at times $t$ and $t + h$ when stationarity is reached. The corresponding characteristic return time $T_c$ to equilibrium is accordingly defined by $\rho(T_c) = 1/e$. We show in appendix C that $\rho(h) = e^{rh}$ (for $r < 0$ corresponding to $R_0 < 1$) in the present model so that $T_c = -1/r$.

The immigration of individuals with some reproductive value determined by their 'age' will have a small effect on the fluctuations of $N - V$, but it is the reproductive value rather than $N$ itself that is the most interesting quantity because $V$ and not $N$ determines the future of the epidemic process. If, for example, immigration is stopped, then the process reverts back to our former 'closed epidemic' model but with the initial value of the process for determining the distribution of the time to extinction being $V_0$ and not $N_0$.

To illustrate the effect of immigration of infection, we have chosen $v_m$ to be the mean value of individuals reproductive values for age classes 1–3 and used an immigration rate $\mu = 1$ corresponding to on average one infected immigrant per day. Figure 1c shows the mean reproductive value of imported cases under this model. We show in figure 3 the expected total reproductive value, which is approximately the number of transmitters, as a function of the reproduction number $R_0$ for the gamma model for infection rates, as well as some illustrations of stationary distributions given by equation (3.1). For other values of $\mu$, the mean and standard deviations shown in the graphs should be multiplied by $\mu$. When restrictions are partly removed to allow some immigration, it is likely that other restrictions also are removed so that the reproductive number $R_0$ comes closer to one. However, the process is stationary only if $R_0 < 1$.

# 4. Computations based on estimates of $R_0$

The purpose of regulations within countries or regions is to make the parameter $R_0$ as small as possible by reducing the mean frequency of close contacts between people. In a closed region we have seen, when $R_0$ is smaller than one, that the number of infectious persons is represented by a decreasing stochastic process eventually reaching extinction with probabilities given by equation (2.1). Accordingly, much effort is used in trying to estimate $R_0$ with as high precision as possible.

However, practically all countries have opened up for some migration, implying that there will be some rate $\mu$ of imported infections. In general, the goal of interventions is, in addition to reducing $R_0$, also to reduce $\mu$. With a constant average rate of imported cases we have seen that there is a stationary

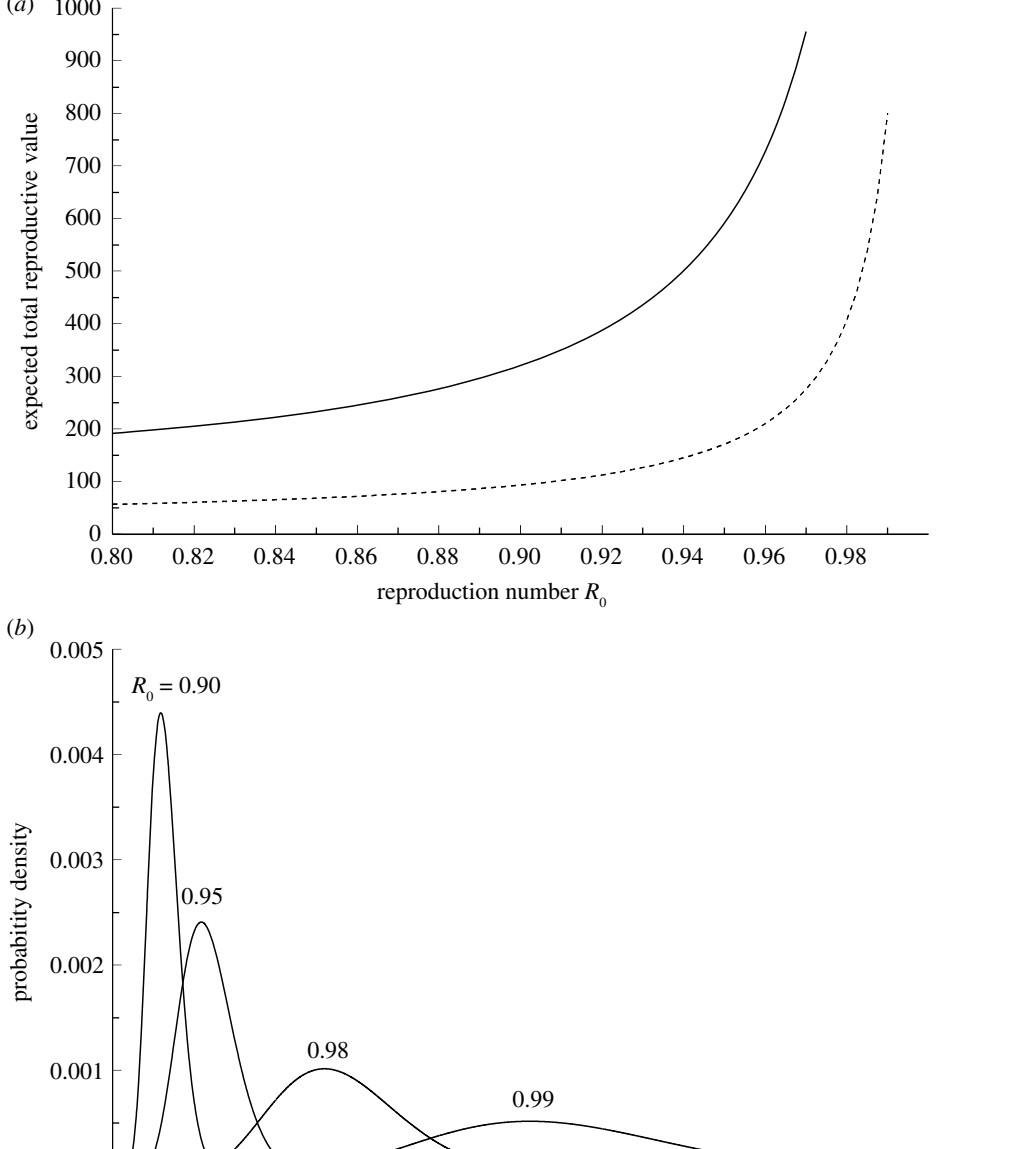

**Figure 3.** The solid line in panel (a) shows the expectation of the stationary distribution of total reproductive value, which is approximately the number infectious persons, as function of the reproduction number $R_0$ for the standard set of parameters used in figure 1 and on average one immigrant per day ($\mu = 1$). For other numbers of immigrants, this expectation must be multiplied by $\mu$. Immigrants are on average assumed to be have been infected in 3 days so that their average reproductive value is $(v_1 + v_2 + v_3)/3$. The dashed line in panel (a) shows the standard deviation s.d.($V$) for $\mu = 1$, and over-dispersal factors $D = D_m = 1$. For other parameters $\mu$ and $D = D_m$, the standard deviations appear by multiplying with $\sqrt{D\mu}$. If $D \neq D_m$ equation (3.2) must be used to find the standard deviation. Panel (b) shows the actual stationary distribution given by equation (3.1) for different values of $R_0$, when the other parameters are as in the upper panel.

distribution of number of transmitters as given by equation (3.1). This distribution has mean $-\mu v_m/r$ where $v_m$ is the average reproductive value of infected immigrants. The variance of the distribution given by equation (3.1) is also proportional to the immigration rate $\mu$. Even if the value of $\mu$ is unknown or very uncertain, it is important to analyse how it affects the number of infections as a guide for management, in particular because the need for hospital capacity is practically proportional to that number.

In figure 1, we have shown three relevant variables that are functions of $R_0$, using the gamma model for infection rates, so that $R_0$ determines the complete Leslie matrix. These are the approximate growth rate $r$ and the value of $v_m$ when immigrants are assumed to be infected less than 4 days so that they are

unlikely to show symptoms. The third variable is the demographic variance $\sigma_1^2$ when transmissions are completely independent and random so that $D = 1$. For other values of $D$ the demographic variance is $\sigma_d^2 = D\sigma_1^2$. Figure 3a shows the expected total reproductive value $EV$ and standard deviation s.d.$(V)$ for $\mu = D = D_m = 1$. For other values of $\mu$ and $D = D_m$ the standard deviation should be multiplied by $\sqrt{D\mu}$, while for $D \neq D_m$ equation (3.2) must be applied.

In order to choose the value of $D$, let us assume that one person has contacts close enough to transmit the virus with probability $p$ to $X$ persons during a day. Writing var$(X) = D_x EX$ for the expectation, the variance in the number of transmissions is $p(1 - p)E\,X + p^2\,D_x E\,X$ giving $D = 1 + (D_x - 1)p$. We see from this that if the number of contacts per day is Poisson distributed so that $D_x = 1$, then also $D = 1$, which is the reference value we have used. One basic goal of interventions is to make $EX$ small, which again makes $R_0$ small since $R_0$ is the sum of the $EX$ during the whole period of transmissions. By contrast, minimizing $D$ is not important when it comes to extinction since, as apparent from our sensitivity analysis, extinctions on average occur sooner as $D$ increases when the epidemic is in a sub-critical regime and there is no immigration.

Over-dispersion in the immigration process relative to the Poisson ($D_m > 1$) will occur if for example immigrants that are infected arrive in groups. If groups arrive at random (Poisson) with a mean and variance of group size $m$ and $\sigma_m^2$, respectively, one can show that the over-dispersion parameter in the immigration term is $D_m = \sigma_m^2/m + m$. Hence, if, for example, group sizes are constant and equal to $m$ then $D_m = m$, so even pairs of infected arriving together will contribute to $D_m$ and the stationary variance given by equation (3.2).

As a further illustration, consider, for example, a country with $R_0 = 0.96$. If, for example, there are 10 infected immigrants on average per day, then the mean and variance for $D = D_m = 1$ are 3515.7 and 71113.0 (standard deviation 267), respectively. If $D = D_m = 4$ the variance must be multiplied by 4 and standard deviation becomes 533.

# 5. Discussion

Our analysis is based on the use of stochastic Leslie theory to infectious disease dynamics. This theory has a long tradition with wide application in ecological research. The main reason for this is that a very complex situation, which needs a large number of parameters to be described in detail, can be studied quantitatively using a model with only three parameters. In the present case when environmental fluctuations are ignored and there is no immigration, there are only two parameters, the multiplicative growth rate $\lambda$, the demographic variance $\sigma_d^2$ (these parameters, in turn, obviously depend on a large number of underlying properties of the biology of the host, the pathogen and their mutual interaction). In the literature on epidemics and recently on the SARS-CoV-2 virus the reproduction number is defined as the average number infected by one person. In a completely susceptible population, this is called the basic reproduction number, $R_0$. In a partially immune population this is called the effective reproduction number $R_E$. Our analysis is also valid for $R_E < R_0$ by replacing $R_0$ by $R_E$ in the calculations. The value of this parameter determines the rise or decline of an epidemic when $R_0 > 1$ or $R_0 < 1$ and is related to the growth rate $\lambda$, but there is not a unique relationship between them unless the 'generation time' (viz. the serial interval) is kept fixed. The $\lambda$ used as a measure of fitness in evolutionary ecology describes how fast the number of infected increases (or decreases). Although $R_0$ by itself does not contain information on the time aspect of the process, it is nevertheless the most important parameter in disease dynamics in, for example, governing the threshold for herd immunity.

Our results are in particular focusing on uncertainty, of which there are mainly two types, parameter and process uncertainty. The uncertainty in parameter values is illustrated in figure 2, and an extended sensitivity analysis can easily be performed for any parameter combinations. Figure 2 also illustrates the process uncertainty through the distribution of time to extinction, while in figure 3b and the variance given by equation (3.2) illustrate the process uncertainty in the case of immigration. These results also show how the process uncertainty depends on the over-dispersion parameters $D$ in transmission and $D_m$ in immigration.

A great advantage in using the diffusion approximation is that it can be applied with few assumptions, only requiring knowledge about the mean and variances of changes. This we have used to go beyond the Poisson assumption allowing for over- or under-dispersion relative to that distribution describing a purely random effect and independence among individuals. The dispersion parameter $D$ (variance to mean ratio) in transmissions simply enters as a factor in $\sigma_d^2$ while the $D_m$ for the immigrations is a factor in the variance term due to immigrations.

The use of fewer variables, the multiplicative growth rate $\lambda$ and the demographic variance $\sigma_d^2$, derived from the properties of the transmission process, can allow more rapid prediction of the shape and size of an epidemic and improved scaling of the response, as the numbers of infected individuals are rapidly changing. Currently, in China, people need to have their 'healthy code' (PCR screening results) for travel between cities or provinces. All international travellers to China receive PCR screening upon their arrival at the airport or at the land border. By this strict screening with PCR tests, China is currently rapidly able to find asymptomatic infections or carriers, which helps to reduce the effective virus migration rate. The results of this effort are visible in the infection data. China has tracked data for numbers of infected persons immigrating from Mars 10, which on average is $\mu = 35.32$ per day. Hence, the values in figure 3*b* should be multiplied with this factor to give an equilibrium value for different values of $R_0$. Assuming as before that imported cases have been infected during the last 3 days, the mean values of the stationary distributions for $R_0 = 0.5, 0.8, 0.98$ are 5652, 6777 and 49 904, respectively. For $D = D_m = 1$, the corresponding standard deviations, are 299, 339 and 2419. Notice that the mean values as well as the standard deviations of these stationary distributions are independent of the number of infectious individuals because there is no immunity reducing the growth at these relatively small numbers of infections. It appears that weakening the interventions, which leads to larger values of $R_0$, may lead to substantial numbers of cases.

Estimates of $R_0$ are important when it comes to interventions, which in terms of our model has two goals, to reduce $R_0$ and $\mu$. From figure 3, it appears that the mean value of the stationary distribution increases very rapidly with increasing $R_0$ when this parameter approaches 1. If $R_0$ is larger than 1 there is no stationary distribution and the epidemic will grow unchecked or until population immunity has been reached. It is interesting in this situation to compare the effects of the two types of regulations, either restrictions within a country to reduce $R_0$, or travel restrictions aiming at reducing the immigration rate $\mu$. To illustrate this, consider a country introducing regulations that reduce $R_0$ from 0.99 to 0.95. Then, the mean value of infectious persons is reduced by a factor 0.21 so that the effect is the same as reducing the mean number of immigrant cases from 100 to 21, while the characteristic return time to equilibrium changes from 662 to 131 days. If the same reduction in $R_0$ is obtained for a country with $R_0 = 0.89$, reaching 0.85, then the reduction is given by a factor 0.75, so that the effect is the same as reducing the number of immigrants from 100 to 75, while the return time goes from 59 to 42 days. As a consequence, in countries with $R_0$ close to 1, a reduction of $R_0$ is an extremely important step towards reducing the number of infectious persons, and hence the need for hospital capacity, and seems to be more important than reduction in $\mu$. On the other hand, if a complete stop of immigration ($\mu = 0$) is a possible alternative, then the mean value is zero. In that case, if $R_0 < 1$, the process will go extinct with probabilities of extinction as a function of time given by equation (2.1) and illustrated by figure 2.

We have not been able to give values of the variance in the stationary distribution, but only report it as a function of the two over-dispersion parameters $D$ and $D_m$ relative to the Poisson distribution. We should expect that both of these parameters are considerably larger than 1. When an infected person transmits the disease, it usually occurs within groups, such as families, public transport units, private parties, or bars. Many countries now apply considerable effort in trying to trace infections by finding all persons who have been in contact with an infected individual. This information can be used to obtain reliable values of $D$ by studying individual variation in transmission during a day. The same applies to the over-dispersion $D_m$ in immigration. When people come into a country, get symptoms and turn out to be infected, one can often trace groups of travellers, test them, and thus find the number of immigrated cases in the group that eventually can be used to estimate $D_m$.

While we have phrased our discussion in terms of the ongoing COVID-19 pandemic, the quantitative approach should be highly relevant to a number of other stage 3 zoonotic diseases [20] with sub-critical human-to-human transmission but immigration from animal reservoirs such as monkey pox, Lhassa fever, hendra and a number of hanta viruses.

Data accessibility. The paper is theoretical, so all values are generated through the model described in the methods.

Authors' contributions. S.E. and N.C.S. conceived and designed the study; S.E. did the theory development and analysis; all authors discussed the results and linked them to observations on the current coronavirus epidemics; S.E., J.D.W. and N.C.S. wrote the paper with input from all authors.

Competing interests. We declare we have no competing interests.

Funding. We received no funding for this study.

Acknowledgements. This work was supported by the authors' respective institutions, especially Centre for Biodiversity Dynamics (CBD), Norwegian University of Science and Technology and Center for Ecological and Evolutionary Synthesis (CEES), University of Oslo. The project was supported by the Research Council of Norway through the COVID-19 Seasonality Project (reference no. 312740) and by the National Key Research and Development Program of China.

# Appendix A. Basic deterministic and stochastic Leslie matrix theory

## A.1. Deterministic theory

In Leslie matrix models used in population dynamics [11,21], the population column vector $n$ with components $n_1, n_2, \ldots, n_k$ represents the number of individuals in each age bracket, with age here interpreted as time since infection. These numbers are propagated forwards in time though the matrix multiplication $Ln$, where $L$ is a Leslie matrix with elements $l_{ij}$ in which the top row represents fecundities $l_{1j}$ and ageing (and survival) is given by the sub-diagonal $l_{(i+1)i}$. In contrast to various previous applications of discrete time age-/stage-structured matrix models in the epidemic context [22], we here think of 'age' as the number of days since a person was infected and the first matrix row as the, possibly, infection-'age'-varying rates of transmission of the virus (as envisioned in the original formulation of the SIR model [23]). Hence, $l_{1j}$ is the average of the rates at which infected persons transmit the virus when they have been infected for $j$ days. For simplicity, we assume that all infectious individuals survive (stay in the population) until they recover and no longer transmit the virus. This can be done for two reasons: first, the death rates during the period considered are rather small, and second, the rates in the first row are mean values so that dead individuals can be considered as being included, but contributing with zeros to the mean.

Leslie matrices have a real dominant eigenvalue $\lambda$ with associated right and left eigenvectors $u$ and $v$ defined by $v^{\mathrm{T}}L = \lambda v^{\mathrm{T}}$ and $Lu = \lambda u$. Here superscript T denotes matrix transposition. After an initial transient period the number of infectious individuals will grow exponentially with multiplicative factor $\lambda$ and approach a stable age distribution. If we scale the eigenvectors by $\sum u_i = 1$ and $\sum u_i v_i = 1$, then $u$ is the stable age distribution and $v$ is the vector of reproductive values for the age classes, a concept introduced by Fisher [10] using a continuous time model. The total reproductive value $V = \sum v_i n_i = v^{\mathrm{T}} n = n^{\mathrm{T}} v$ has exactly exponential growth with multiplicative rate $\lambda$ [10,14] while the total actual number of infectious individuals $N = \sum n_i$ will have minor transient fluctuations around $V$ and is equal to $V$ if $n = Nu$. This can easily be seen by evaluating the change in $V$ during a time step. With subscript $t$ indicating time, we find that $V_{t+1} = v^{\mathrm{T}} n_{t+1} = v^{\mathrm{T}} L n_t = \lambda v^{\mathrm{T}} n_t = \lambda V_t$. If the number of infectious individuals has the stable 'age' distribution then the total reproductive value equals the actual number of infectious individuals since, if $n = Nu$, then $V = v^{\mathrm{T}} n = N v^{\mathrm{T}} u = N$.

Epidemics are commonly studied through the parameter $R_0$, which is the expected total number of infections by a single infected person. Accordingly, the relation to the Leslie matrix theory is that $R_0 = \sum_j l_{1j}$. If $R_0$ (in ecology called lifetime reproductive success) equals one, the $\lambda = 1$, while values greater or smaller than one corresponds to $\lambda$ being greater or smaller than one. If some fraction of the population is immune, then the elements $l_{1j}$ will be smaller and their sum then corresponds to the effective reproduction number $R_E$.

## A.2. Stochastic theory

In stochastic Leslie theory, the elements of the matrix are stochastic variables. From the above results in deterministic theory, one can show that under stochastic temporal variation in the Leslie matrix with no temporal autocorrelations, the total reproductive value $V$ will approximately be a process with white noise, while $N$ is a more complicated process with $N - V$ fluctuating around zero. Hence, there are advantages working with $V$ rather than $N$ since also, it is the value of $V$ that contains the relevant information on the future of the process.

Generally, there may be both environmental noise generated by environmental conditions varying between time steps and demographic noise, which is independent among individuals [14]. Temporal environmental fluctuations are not expected to have much effect on an epidemic, so we only consider demographic stochasticity. Stochastic simulations have confirmed that the process $V$ is well approximated by a diffusion [24] with infinitesimal mean and variance $rV$ and $\sigma_d^2 V$, respectively, where $r = \lambda - 1$, $\sigma_d^2$ is the demographic variance of the process [15,16], and the environmental noise is ignored. This variance is expressed through the individual reproductive values, which are each individual's contribution to the total reproductive value in the next time step [16]. If an individual of age $j$ produces $B$ offspring in a time step, corresponding to infecting $B$ persons, and $I = 1$ if it survives and otherwise zero, then its individual reproductive value is $W_j = v_{j+1} I + v_1 B$. With this notation the demographic variance for the process $V$ is defined as $\sigma_d^2 = \sum u_j \mathrm{var}(W_j)$. In the present model, the demographic stochasticity is assumed only to occur in the transmission process, the 'fecundities', since $I = 1$ and thus has no stochasticity. A reasonable null-model is that transmissions are purely random

so that the numbers infected by individuals during a time step are Poisson distributed with means $l_{1j}$, and hence have variance equal to the mean. Modelling transmissions in continuous time using exponential distributed waiting times resulting in Poisson-distributed number of transmissions during a time interval is common in epidemiology. The Poisson assumption may be seen as a large-population approximation to the chain-binomial epidemic model [17] and is also the assumption employed in $\tau$-leap simulation of event-based epidemics (e.g. [19]).

Accounting for over- or under-dispersion relative to the Poisson, which may for example arise when considering an epidemic as a stochastic birth-and-death process [25], we write $Dl_{1j}$ for the variances. Since there is no stochasticity in the sub-diagonal elements, the variance of the individual reproductive values are $\text{var}(W_j) = Dl_{1j}v_1^2$ because transmissions produce new members of the group of infectious individuals with reproductive value $v_1$. Hence, using the fact that $\sum u_j l_{1j} = \lambda u_1$ we see that $\sigma_d^2 = D\sigma_1^2 = D\lambda v_1^2 u_1$, where $\sigma_1^2 = \lambda v_1^2 u_1$ is the demographic variance under the assumption that new cases during a day is Poisson distributed.

# Appendix B. The cumulant generating function for the stationary distribution

We consider diffusions $V$ with infinitesimal mean $E\,[dV \mid V = v]/dt = -\alpha v + \beta$ and $E\,[(dV)^2 \mid V = v]/dt = \gamma v + \delta$, where $\alpha$, $\beta$, $\gamma$ and $\delta$ are positive constants. Let $V$ and $V + dV$ be the state of the process at time $t$ and $t + dt$, respectively. The moment generating function of $V$ is $M_t(u) = E[e^{uV}]$ while $K_t(u) = \ln M_t(u)$ is the corresponding cumulant generating function [26]. A balance equation then expresses that the distribution of $V$ and $V + dV$ are the same when the stationarity is reached, that is $M_t(u) = M_{t+dt}(u)$. By definition, we have

$$M_{t+dt}(u) = E[e^{u(V+dV)}] = E[E\,e^{V+dV}|V].$$

Using that $E[e^{u\,dV}|V = v] = e^{(-\alpha v + \beta)u\,dt + (\gamma v + \delta)u^2\,dt/2}$ this gives

$$M_{t+dt}(u) = E\,e^{uV + (-\alpha V + \beta)u\,dt + (\gamma V + \delta)u^2\,dt/2} = M_t[u + (-\alpha u + \gamma u^2/2)\,dt]\,e^{\beta u + \delta u^2\,dt/2}.$$

Taking the logarithm of each side, this yields

$$K_{t+dt}(u) = K_t\left[u + \left(-\alpha u + \frac{\gamma u^2}{2}\right)dt\right] + \left(\beta u + \frac{\delta u^2}{2}\right)dt,$$

which gives the balance equation at stationarity as

$$K_t(u) = K_{t+dt}(u) = K_t(u) + K_t'(u)\left(-\alpha u + \frac{\gamma u^2}{2}\right)dt + \left(\beta u + \frac{\delta u^2}{2}\right)dt$$

and the differential equation, omitting the time subscript at stationarity,

$$K'(u) = -\frac{\beta + \delta u/2}{-\alpha + \gamma u/2} = -\frac{\delta}{\gamma} + \frac{2}{\gamma}\left(\beta + \frac{\alpha\delta}{\gamma}\right)\left(-\frac{2\alpha}{\gamma} + u\right)^{-1}.$$

From this, we find all cumulants $\kappa_n$ for $n = 2, 3, \ldots$ as the $n'$th derivative of $K(u)$ at $u = 0$,

$$\kappa_n = \frac{2}{\gamma}\left(\beta + \frac{\alpha\delta}{\gamma}\right)(-1)^n(n-1)!\left(\frac{\gamma}{2\alpha}\right)^n,$$

while $\kappa_1 = \beta/\alpha$. This yields the mean and variance of the stationary distribution as

$$EV = K'(0) = \kappa_1 = \frac{\beta}{\alpha}, \quad \text{var}(V) = K''(0) = \kappa_2 = \left(\frac{\delta}{\beta} + \frac{\gamma}{\alpha}\right)\frac{EV}{2}.$$

The skewness and kurtosis of the distribution are $\kappa_3/\kappa_2^{3/2}$ and $\kappa_4/\kappa_2^2$, respectively.

For the model in the main text, we have $\alpha = -r$, $\beta = v_m\mu$, $\gamma = \sigma_d^2 = \sigma_1^2 D$ and $\delta = v_m^2 D_m\mu$, confirming the results given in the main text

$$EV = -\frac{v_m\mu}{r}, \quad \text{var}(V) = \left(-\frac{\sigma_1^2 D}{r} + v_m D_m\right)EV/2.$$

# Appendix C. Temporal autocorrelation

Consider again the process with state $V_t$ at time $t$ described by the infinitesimal mean and variance $-\alpha v + \beta$ and $\nu(v)$, respectively, where $\alpha > 0$. When stationarity is reached the temporal autocovariance function is $c(h) = \mathrm{cov}(V_t, V_{t+h})$. Then, $dV_{t+h} = (-\alpha V_{t+h} + \beta)\mathrm{d}t + \sqrt{\nu(V_{t+h})}\mathrm{d}B(t + h)$, where $dB(t + h)$ is the increment of a standard Brownian motion during the time interval from $t + h$ to $t + h + dh$. with $E dB(t + h) = 0$ and $\mathrm{var}[dB(t + h)] = dh$. This gives

$$c(h + dh) = \mathrm{cov}(V_t, V_{t+h} + \mathrm{d}V_{t+h}).$$

We can then argue conditionally on $V_{t+h}$ and using the well-known result that $\mathrm{cov}(V_t, \sqrt{V_{t+h}}\,dB(t + h))$ is the expectation of this covariance conditioned on $V_{t+h}$ plus the covariance between the expected values of $V_t$ and the increment $\sqrt{V_{t+h}}\,dB(t + h)$ when these are both conditioned on $V_{t+h}$. Since $dB(t + h)$ by definition is independent of the past, both of these terms are zero so that the unconditional covariance is also zero. Accordingly

$$c(h + \mathrm{d}h) = \mathrm{cov}[V_t, V_{t+h}(1 - \alpha\,\mathrm{d}h] = c(h)(1 - \alpha\,\mathrm{d}h),$$

giving $c'(h) = -\alpha c(h)$. Hence, $c(h) = c(0)\,\mathrm{e}^{-\alpha h}$, and the corresponding temporal autocorrelation is simply $\rho(h) = \mathrm{e}^{-\alpha h}$.

For the model in the main text, we have $\alpha = -r$ so that the temporal autocorrelation function is $\rho(h) = \mathrm{e}^{rh}$.

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
