## [Peer Review File · Royal Society Open Science]

Review History

RSOS-202234.R0 (Original submission)

Review form: Reviewer 1

Is the manuscript scientifically sound in its present form?

Yes

Are the interpretations and conclusions justified by the results?

Yes

Is the language acceptable?

Yes

Do you have any ethical concerns with this paper?

No

Have you any concerns about statistical analyses in this paper?

No

Recommendation?

Major revision is needed (please make suggestions in comments)

Comments to the Author(s)

This is a really interesting and important paper with crucial results for when we reach the stage when $R_0 < 1$ for Covid-19 in most communities. I strongly believe this paper should be published, but I think it would have much stronger impact if details of the technical analyses were moved to an Appendix and the authors focus on describing in more detail the implications of their results.

I also wonder if it is wise to base much of the discussion around R_0 , most of the current attention is around R_{eff} , so I think it might be good to rephrase the manuscript to reflect this.

Of particular importance is that I'd like to see more discussion of time to extinction when R_{eff} is only just less than unity. I suspect that the global public has been left with the impression that all be well when $R < 1$, but it might take a long time to reach eradication when $R \sim 0.9$ or even 0.8, the changes in behaviour when $R < 1$, have led to very rapid bouncebacks.

It might also be good to mention that the emergence of more transmissible strains make it harder to reduce $R < 1$ and even harder to levels where eradication is on the near horizon.

Review form: Reviewer 2

Is the manuscript scientifically sound in its present form?

No

Are the interpretations and conclusions justified by the results?

No

Is the language acceptable?

Yes

Do you have any ethical concerns with this paper?

No

Have you any concerns about statistical analyses in this paper?

Yes

Recommendation?

Major revision is needed (please make suggestions in comments)

Comments to the Author(s)

This manuscript presents the application of Leslie's theory, originally developed in ecology for population dynamics, to describe the transmission of a disease in a country during a recession phase ($R_0 < 1$), with the particular application to COVID-19. Results mainly show a sensitivity analysis of the reproduction number and time to extinction when taking into consideration stochastic parameters in Leslie's matrix and possible immigration of infected individuals. TO these more theoretical results, the authors add estimates of R_0 for some countries at the beginning of August 2020, together with the estimated mean number of infections caused by immigrants.

Although the topic is of high relevance and actuality, my opinion is that the manuscript, in the present form, does not satisfy the high quality standards of this journal. In particular, the method section needs to be deeply rewritten, using a better notation and clearly linking the variables to the epidemiological model. The described approach is quite general and could be applied to different diseases, however, I find that the set up of the model for the application to COVID-19 is very weak, and should rely on data and results from the literature (for example, the model could replicate the generation times found in other publications).

Major comments

- 1) Model section: the description of the model needs a deep re-writing: the authors make a nice parallel between Leslie model and the epidemiological model (beginning of page 7), but this parallel should be before in the model section. All the details about the eigenvectors/eigenvalues of the matrix and the introduction of the stochasticity should be better explained, but always referring to the epidemiological model.
- 2) One of my major concerns is that the model does not consider that the number of susceptible individuals decreases during the epidemic and, thus, the transmission rates in Leslie matrix should change. Since in a country the population is not infinite, the Authors should take into consideration the depletion of susceptible into the model.
- 3) Evidence from COVID-19 patients shows that the generation times have a humped shape distribution, typically model as a gamma function (see, e.g., Du et al. 2020, Cereda et al. 2020). Having age classes for the infectious period would be very helpful to describe these generation times. However, the authors assume a latency period with 0 transmission followed by a constant transmission rate (page 8, lines 8-13). I suggest changing these assumptions and impose transmission rates in agreement with a gamma function.
- 4) It is very difficult to understand how the estimates of R_t in table 1 have been computed and which data have been used. Moreover, the outbreaks in the different countries have been characterized by very different transmission rates due, for example, to different policies on social distancing. I find unrealistic the application of the same formula with the same parameters to describe the epidemic in very different countries. For example, the R_t indicated for some countries in table 1 differ from the ones computed in the website <https://covid19-projections.com/>

Du Z., Xu X., Wu Y. Serial interval of COVID-19 among publicly reported confirmed cases. *Emerg. Infect. Dis.* 2020;26:1341–1343. doi: 10.3201/eid2606.200357.

Cereda D., Tirani M., Rovida F. ArXiv; Italy: 2020. The Early Phase of the COVID-19 Outbreak in Lombardy. <https://arxiv.org/abs/2003.09320>

MINOR COMMENTS

Significant statement: The significant statement is usually not present in article of the Royal Society Open Science and it is partially a repetition of the abstract. Please merge this part with the abstract respecting the limit on the word count.

Page 3, line 27: the acronym NPI is used only here.

Page 3, line 29: 'The implementation of these to control measures has been' please revise the sentence.

Page 4, line 18: 'post-pandemic era': here it would be necessary to insert a discussion on the mass vaccination campaigns that are ongoing worldwide

Page 4, line 27: 'transmission dynamics that we may see after...' in reality the consequences of decreasing social distancing have been already evident for example in Europe, with the surge of cases after summer.

Page 4, line 46-48: how are quarantine measures for immigrant individuals taken into consideration in this paper? For example, Italy was imposing a quarantine during summer to most people entering the country.

Page 4, line 52-54: 'using an ecological approach'. I suggest rephrasing in 'using an approach largely adopted in ecological studies'.

Page 5, line 28: please clarify what quantity follows an inverse Gaussian distribution.

Page 5, line 41: REF?

Page 6, line 16: I suggest writing explicitly the dependence on time and the main equation of the model $N(t_k) = L * N(t_{(k-1)})$.

Page 6, line 27-28: please insert here the more detailed description of the model that is now at the beginning of page 7. The assumption that no individual dies during the infection is very strong and probably not applicable to COVID-19. Moreover infected individuals might be hospitalized or isolated in quarantine or recovered at different times, thus not contributing to the infection. Is it possible to take into consideration these aspects into the model?

Page 6, lines 30-40: please review this paragraph on the eigenvalues/eigenvectors highlighting the meaning of u and v in the description of the epidemic.

Page 6, line 35: the product among vectors vn should be $v' n$.

Page 6, line 43: also this part on the stochastic model should be revised to better describe how stochasticity is implemented in the case of the epidemiological model. What are the environmental and the demographic noise in this case?

Page 6, line 51: the notation is not clear. What is r (or rV)?

Page 7, line 36: these are the variances of what?

Page 7, line 43: 'the new cases during a day is Poisson distributed'. Please replace 'is' with 'are'. It is not clear how over/under dispersions are considered in the Poisson distribution (which has only one parameter). Are the authors using a negative binomial distribution?

Page 7, eq(1): how does this equation changes for the model without stochasticity? If taking into consideration that the number of susceptible is finite, is this formula representing an upper limit for the extinction? I think that n_0 at line 52 should be N_0 .

Page 8, lines 8-22: this part is probably the core of the epidemiological model proposed and I suggest to move it before in the methods section and add some more details. For example, it must be clear that columns in the Leslie matrix correspond days after the transmission. The period of infectiousness typically changes among individuals. Does the stochastic model take this aspect into consideration?

Page 8, line 24: are the authors referring to eq. 1? D does not appear explicitly in this equation, thus I suggest replacing σ_d by its relationship with D if possible. Since D is the only factor accounting for the stochasticity, I suggest to better describe its role in the model.

Page 8, line 29 in the legend of fig. 1 there is the variable D_m that has not been mentioned up to now.

Page 8, lines 39-41: this assumption should be clarified earlier, when presenting the model.

Page 8, lines 48-50: the authors say that they assume that the immigrants have been infected recently. Does this mean that all immigrants are assume to enter the first infectious age? Please add details.

Page 9, line 28: replace 'has' with 'have'

Page 9, eq. (2): from the text I do not understand what this equation represents. From the legend of Fig. 2, I can understand that it is related to the distribution of the total reproductive value, and thus to the distribution of the infected cases at equilibrium. But this is not clear in the text. Moreover, the system at equilibrium should have zero infected since the epidemic should be extinct.

Page 9, line 42: The equilibrium for this system should be the extinction of the epidemic, but I am not sure if this is the equilibrium that is mentioned here. Please clarify. In this case, are the fluctuations due to the immigration process?

Table 1: how are these estimates of R_t obtained? Is it reasonable to assume the same model parameters for all countries, where different policies were imposed? Please mention the source of the data used and add details about the data? The estimation of R_t is a difficult task when referring to one country, thus it is very difficult to understand how reliable are these estimates for

multiple countries. Please note that in the main text the authors speak about R_0 , but in the table there is R_t .

Page 10, line 15: what is R_{eff} ?

Page 12, line 15-16: what are variables D_x and E ? these details would be interesting if better explained. Probably this part should go in the methods

Decision letter (RSOS-202234.R0)

Dear Dr Whittington

The Editors assigned to your paper RSOS-202234 "The ecological dynamics of the Corona epidemics during transmission from outside sources when R_0 is successfully managed below one" have now received comments from reviewers and would like you to revise the paper in accordance with the reviewer comments and any comments from the Editors. Please note this decision does not guarantee eventual acceptance.

Please submit your revised manuscript and required files (see below) no later than 21 days from today's (ie 11-Feb-2021) date. Note: the ScholarOne system will 'lock' if submission of the revision is attempted 21 or more days after the deadline. If you do not think you will be able to meet this deadline please contact the editorial office immediately.

on behalf of Professor Enrico Bertuzzo (Associate Editor) and Pete Smith (Subject Editor)
 openscience@royalsociety.org

Associate Editor Comments to Author (Professor Enrico Bertuzzo):

Comments to the Author:

I apologized for the long time it took to finalize the first round of reviews, but it has been rather challenging to secure reviewers during the holiday break. At any rate, the manuscript has now been reviewed by two experts in the field. They both find that the methods developed are interesting and original, but that a major revision would be needed to properly present the material. I share their opinion. In particular, the authors should keep in mind the general audience RSOS is aimed at when revising their manuscript. For instance, a graphical illustration of a Leslie matrix would be helpful to accompany the description in page 7. The reviewers offer detailed suggestions for the revision that I am confident the authors will consider.

Reviewer comments to Author:

Reviewer: 1

Comments to the Author(s)

This is a really interesting and important paper with crucial results for when we reach the stage when $R_0 < 1$ for Covid-19 in most communities. I strongly believe this paper should be published, but I think it would have much stronger impact if details of the technical analyses were moved to an Appendix and the authors focus on describing in more detail the implications of their results.

I also wonder if it is wise to base much of the discussion around R_0 , most of the current attention is around R_{eff} , so I think it might be good to rephrase the manuscript to reflect this.

Of particular importance is that I'd like to see more discussion of time to extinction when R_{eff} is only just less than unity. I suspect that the global public has been left with the impression that all be well when $R < 1$, but it might take a long time to reach eradication when $R \sim 0.9$ or even 0.8, the changes in behaviour when $R < 1$, have led to very rapid bouncebacks.

It might also be good to mention that the emergence of more transmissible strains make it harder to reduce $R < 1$ and even harder to levels where eradication is on the near horizon.

Reviewer: 2

Comments to the Author(s)

This manuscript presents the application of Leslie's theory, originally developed in ecology for population dynamics, to describe the transmission of a disease in a country during a recession phase ($R_0 < 1$), with the particular application to COVID-19. Results mainly show a sensitivity analysis of the reproduction number and time to extinction when taking into consideration stochastic parameters in Leslie's matrix and possible immigration of infected individuals. TO these more theoretical results, the authors add estimates of R_0 for some countries at the beginning of August 2020, together with the estimated mean number of infections caused by immigrants. Although the topic is of high relevance and actuality, my opinion is that the manuscript, in the present form, does not satisfies the high quality standards of this journal. In particular, the method section needs to be deeply rewritten, using a better notation and clearly linking the variables to the epidemiological model. The described approach is quite general and could be applied to different diseases, however, I find that the set up of the model for the application to COVID-19 is very weak, and should rely on data and results from the literature (for example, the model could replicate the generation times found in other publications).

Major comments

- 1) Model section: the description of the model needs a deep re-writing: the authors make a nice parallel between Leslie model and the epidemiological model (beginning of page 7), but this parallel should be before in the model section. All the details about the eigenvectors/eigenvalues of the matrix and the introduction of the stochasticity should be better explained, but always referring to the epidemiological model.
- 2) One of my major concerns is that the model does not consider that the number of susceptible individuals decreases during the epidemic and, thus, the transmission rates in Leslie matrix should change. Since in a country the population is not infinite, the Authors should take into consideration the depletion of susceptible into the model.
- 3) Evidence from covid19 patience shows that the generation times have a humped shape distribution, typically model as a gamma function (see, e.g., Du et al. 2020, Cereda et al. 2020). Having age classes for the infectious period would be very helpful to describe these generation times. However, the authors assume a latency period with 0 transmission followed by a constant transmission rate (page 8, lines 8-13). I suggest changing these assumptions and impose transmission rates in agreement with a gamma function.
- 4) It is very difficult to understand how the estimates of R_t in table 1 have been computed and which data have been used. Moreover, the outbreaks in the different countries have been characterized by very different transmission rates due, for example, to different policies on social distancing. I find unrealistic the application of the same formula with the same parameters to describe the epidemic in very different countries. For example, the R_t indicated for some countries in table 1 differ from the ones computed in the website <https://covid19-projections.com/>

Du Z., Xu X., Wu Y. Serial interval of COVID-19 among publicly reported confirmed cases. *Emerg. Infect. Dis.* 2020;26:1341-1343. doi: 10.3201/eid2606.200357.

Cereda D., Tirani M., Rovida F. ArXiv; Italy: 2020. The Early Phase of the COVID-19 Outbreak in Lombardy. <https://arxiv.org/abs/2003.09320>

MINOR COMMENTS

Significant statement: The significant statement is usually not present in article of the Royal Society Open Science and it is partially a repetition of the abstract. Please merge this part with the abstract respecting the limit on the word count.

Page 3, line 27: the acronym NPI is used only here.

Page 3, line 29: 'The implementation of these to control measures has been' please revise the sentence.

Page 4, line 18: 'post-pandemic era': here it would be necessary to insert a discussion on the mass vaccination campaigns that are ongoing worldwide

Page 4, line 27: 'transmission dynamics that we may see after...' in reality the consequences of decreasing social distancing have been already evident for example in Europe, with the surge of cases after summer.

Page 4, line 46-48: how are quarantine measures for immigrant individuals taken into consideration in this paper? For example, Italy was imposing a quarantine during summer to most people entering the country.

Page 4, line 52-54: 'using an ecological approach'. I suggest rephrasing in 'using an approach largely adopted in ecological studies'.

Page 5, line 28: please clarify what quantity follows an inverse Gaussian distribution.

Page 5, line 41: REF?

Page 6, line 16: I suggest writing explicitly the dependence on time and the main equation of the model $N(t_k) = L * N(t_{(k-1)})$.

Page 6, line 27-28: please insert here the more detailed description of the model that is now at the beginning of page 7. The assumption that no individual dies during the infection is very strong and probably not applicable to COVID-19. Moreover infected individuals might be hospitalized

or isolated in quarantine or recovered at different times, thus not contributing to the infection. Is it possible to take into consideration these aspects into the model?

Page 6, lines 30-40: please review this paragraph on the eigenvalues/eigenvectors highlighting the meaning of u and v in the description of the epidemic.

Page 6, line 35: the product among vectors vn should be $v' n$.

Page 6, line 43: also this part on the stochastic model should be revised to better describe how stochasticity is implemented in the case of the epidemiological model. What are the environmental and the demographic noise in this case?

Page 6, line 51: the notation is not clear. What is r (or rV)?

Page 7, line 36: these are the variances of what?

Page 7, line 43: 'the new cases during a day is Poisson distributed'. Please replace 'is' with 'are'. It is not clear how over/under dispersions are considered in the Poisson distribution (which has only one parameter). Are the authors using a negative binomial distribution?

Page 7, eq(1): how does this equation changes for the model without stochasticity? If taking into consideration that the number of susceptible is finite, is this formula representing an upper limit for the extinction? I think that n_0 at line 52 should be N_0 .

Page 8, lines 8-22: this part is probably the core of the epidemiological model proposed and I suggest to move it before in the methods section and add some more details. For example, it must be clear that columns in the Leslie matrix correspond days after the transmission. The period of infectiousness typically changes among individuals. Does the stochastic model take this aspect into consideration?

Page 8, line 24: are the authors referring to eq. 1? D does not appear explicitly in this equation, thus I suggest replacing σ_d by its relationship with D if possible. Since D is the only factor accounting for the stochasticity, I suggest to better describe its role in the model.

Page 8, line 29 in the legend of fig. 1 there is the variable D_m that has not been mentioned up to now.

Page 8, lines 39-41: this assumption should be clarified earlier, when presenting the model.

Page 8, lines 48-50: the authors say that they assume that the immigrants have been infected recently. Does this mean that all immigrants are assume to enter the first infectious age? Please add details.

Page 9, line 28: replace 'has' with 'have'

Page 9, eq. (2): from the text I do not understand what this equation represents. From the legend of Fig. 2, I can understand that it is related to the distribution of the total reproductive value, and thus to the distribution of the infected cases at equilibrium. But this is not clear in the text. Moreover, the system at equilibrium should have zero infected since the epidemic should be extinct.

Page 9, line 42: The equilibrium for this system should be the extinction of the epidemic, but I am not sure if this is the equilibrium that is mentioned here. Please clarify. In this case, are the fluctuations due to the immigration process?

Table 1: how are these estimates of R_t obtained? Is it reasonable to assume the same model parameters for all countries, where different policies were imposed? Please mention the source of the data used and add details about the data? The estimation of R_t is a difficult task when referring to one country, thus it is very difficult to understand how reliable are these estimates for multiple countries. Please note that in the main text the authors speak about R_0 , but in the table there is R_t .

Page 10, line 15: what is R_{eff} ?

Page 12, line 15-16: what are variables D_x and E ? these details would be interesting if better explained. Probably this part should go in the methods

===PREPARING YOUR MANUSCRIPT===

===PREPARING YOUR REVISION IN SCHOLARONE===

- An editable file of each table (.doc, .docx, .xls, .xlsx, or .csv).
- An editable file of all figure and table captions.

- Any electronic supplementary material (ESM).
- If you are requesting a discretionary waiver for the article processing charge, the waiver form must be included at this step.
- If you are providing image files for potential cover images, please upload these at this step, and inform the editorial office you have done so. You must hold the copyright to any image provided.
- A copy of your point-by-point response to referees and Editors. This will expedite the preparation of your proof.

- Ensure that your data access statement meets the requirements at <https://royalsociety.org/journals/authors/author-guidelines/#data>. You should ensure that you cite the dataset in your reference list. If you have deposited data etc in the Dryad repository, please include both the 'For publication' link and 'For review' link at this stage.
- If you are requesting an article processing charge waiver, you must select the relevant waiver option (if requesting a discretionary waiver, the form should have been uploaded at Step 3 'File upload' above).
- If you have uploaded ESM files, please ensure you follow the guidance at <https://royalsociety.org/journals/authors/author-guidelines/#supplementary-material> to include a suitable title and informative caption. An example of appropriate titling and captioning may be found at https://figshare.com/articles/Table_S2_from_Is_there_a_trade-off_between_peak_performance_and_performance_breadth_across_temperatures_for_aerobic_scope_in_teleost_fishes_/3843624.

Author's Response to Decision Letter for (RSOS-202234.R0)

See Appendix A.

RSOS-202234.R1 (Revision)

Review form: Reviewer 2

Is the manuscript scientifically sound in its present form?

Yes

Are the interpretations and conclusions justified by the results?

Yes

Is the language acceptable?

Yes

Do you have any ethical concerns with this paper?

No

Have you any concerns about statistical analyses in this paper?

No

Recommendation?

Accept with minor revision (please list in comments)

Comments to the Author(s)

The authors replied in detail to all my comments. I find that the manuscript has been improved, the method section is now quite clear and easy to read. I only have few additional minor comments.

One thing that I find a bit confusing is that the authors frequently use the term 'population' or 'total population' to refer to the number of infectious individuals. I imagine this term is normal when using the Leslie matrix for ecological models. However, in epidemiology the term population usually refers to the total population of the region or country under exam, which in this model is not even taken into consideration. This might generate some confusion, especially when introducing crucial variables such as V . I suggest to carefully re-read the manuscript and, when necessary, replace the term 'population' with 'number of infectious individuals'.

Please find in the following other minor comments.

Line 119-122: articles are missing

Line 122: please clarify the sentence 'survivals up to age when there are no longer transmission are 1'.

Line 128: if I understand correctly, each column of the Leslie matrix corresponds to a time step, which in this model is a day. I think it is better to specify this here. Thus the rates are daily rates, correct?

Line 131: please specify 'The number of transmitters'

Line 135: why the sum of the components of the stable age distribution is one? In the previous line the authors stated that the population grows approximately exponentially! I guess there should be a multiplication by a scalar (which depends on time) in front of the eigenvector u . Please clarify.

Line 164: please define n_0 .

Line 177: Here the focus is the stationary distribution of V . However, I still have to completely understand what it represents from an epidemiological point of view. In lines 146-152 I find the definition of V , but I do not see why for immigration our focus is this stationary distribution.

Line 430: 'total reproductive'. Probably missing 'number'

Line 595, Fig1, legend: In panel 1 of figure 1 there is not a log-scale. Please correct the legend and revise also line 248

Decision letter (RSOS-202234.R1)

Dear Dr Whittington

On behalf of the Editors, we are pleased to inform you that your Manuscript RSOS-202234.R1 "The ecological dynamics of the Corona epidemics during transmission from outside sources when R_0 is successfully managed below one" has been accepted for publication in Royal Society Open Science subject to minor revision in accordance with the referees' reports. Please find the referees' comments along with any feedback from the Editors below my signature.

Please submit your revised manuscript and required files (see below) no later than 7 days from today's (ie 06-May-2021) date. Note: the ScholarOne system will 'lock' if submission of the revision is attempted 7 or more days after the deadline. If you do not think you will be able to meet this deadline please contact the editorial office immediately.

on behalf of Professor Enrico Bertuzzo (Associate Editor) and Pete Smith (Subject Editor)
openscience@royalsociety.org

Associate Editor Comments to Author (Professor Enrico Bertuzzo):
Comments to the Author:

The manuscript has been reviewed by the two original reviewers and both are satisfied with the revision (one review report is not included because the reviewers simply replied by mail with a positive feedback). The first reviewer points to some minor corrections that I am confident the authors can implement while preparing the final files.

Reviewer comments to Author:

Reviewer: 2

Comments to the Author(s)

The authors replied in detail to all my comments. I find that the manuscript has been improved, the method section is now quite clear and easy to read. I only have few additional minor comments.

One thing that I find a bit confusing is that the authors frequently use the term 'population' or 'total population' to refer to the number of infectious individuals. I imagine this term is normal when using the Leslie matrix for ecological models. However, in epidemiology the term population usually refers to the total population of the region or country under exam, which in this model is not even taken into consideration. This might generate some confusion, especially when introducing crucial variables such as V . I suggest to carefully re-read the manuscript and, when necessary, replace the term 'population' with 'number of infectious individuals'.

Please find in the following other minor comments.

Line 119-122: articles are missing

Line 122: please clarify the sentence 'survivals up to age when there are no longer transmission are 1'.

Line 128: if I understand correctly, each column of the Leslie matrix corresponds to a time step, which in this model is a day. I think it is better to specify this here. Thus the rates are daily rates, correct?

Line 131: please specify 'The number of transmitters'

Line 135: why the sum of the components of the stable age distribution is one? In the previous line the authors stated that the population grows approximately exponentially! I guess there should be a multiplication by a scalar (which depends on time) in front of the eigenvector u . Please clarify.

Line 164: please define n_0 .

Line 177: Here the focus is the stationary distribution of V . However, I still have to completely understand what it represents from an epidemiological point of view. In lines 146-152 I find the definition of V , but I do not see why for immigration our focus is this stationary distribution.

Line 430: 'total reproductive'. Probably missing 'number'

Line 595, Fig1, legend: In panel 1 of figure 1 there is not a log-scale. Please correct the legend and revise also line 248

===PREPARING YOUR MANUSCRIPT===

Your revised paper should include the changes requested by the referees and Editors of your manuscript. You should provide two versions of this manuscript and both versions must be provided in an editable format:
one version identifying all the changes that have been made (for instance, in coloured highlight, in bold text, or tracked changes);

===PREPARING YOUR REVISION IN SCHOLARONE===

- If you are requesting a discretionary waiver for the article processing charge, the waiver form must be included at this step.
- If you are providing image files for potential cover images, please upload these at this step, and inform the editorial office you have done so. You must hold the copyright to any image provided.
- A copy of your point-by-point response to referees and Editors. This will expedite the preparation of your proof.

- Ensure that your data access statement meets the requirements at <https://royalsociety.org/journals/authors/author-guidelines/#data>. You should ensure that you cite the dataset in your reference list. If you have deposited data etc in the Dryad repository, please only include the 'For publication' link at this stage. You should remove the 'For review' link.
- If you are requesting an article processing charge waiver, you must select the relevant waiver option (if requesting a discretionary waiver, the form should have been uploaded at Step 3 'File upload' above).
- If you have uploaded ESM files, please ensure you follow the guidance at <https://royalsociety.org/journals/authors/author-guidelines/#supplementary-material> to include a suitable title and informative caption. An example of appropriate titling and captioning may be found at https://figshare.com/articles/Table_S2_from_Is_there_a_trade-off_between_peak_performance_and_performance_breadth_across_temperatures_for_aerobic_scope_in_teleost_fishes_/3843624.

Author's Response to Decision Letter for (RSOS-202234.R1)

See Appendix B.

Decision letter (RSOS-202234.R2)

Dear Dr Whittington,

I am pleased to inform you that your manuscript entitled "The ecological dynamics of the coronavirus epidemics during transmission from outside sources when R_0 is successfully managed below one" is now accepted for publication in Royal Society Open Science.

COVID-19 rapid publication process:

We are taking steps to expedite the publication of research relevant to the pandemic. If you wish, you can opt to have your paper published as soon as it is ready, rather than waiting for it to be published the scheduled Wednesday.

This means your paper will not be included in the weekly media round-up which the Society sends to journalists ahead of publication. However, it will still appear in the COVID-19 Publishing Collection which journalists will be directed to each week (<https://royalsocietypublishing.org/topic/special-collections/novel-coronavirus-outbreak>).

If you wish to have your paper considered for immediate publication, or to discuss further, please notify openscience_proofs@royalsociety.org and press@royalsociety.org when you respond to this email.

on behalf of Professor Enrico Bertuzzo (Associate Editor) and Pete Smith (Subject Editor)
openscience@royalsociety.org

Appendix A

Dear Professor Enrico Bertuzzo (Associate Editor), Pete Smith (Subject Editor) and Anita Kristiansen, Editorial Coordinator,

Manuscript ID RSOS-202234

Thank you very much for your feedback of February 11 this year on our submission RSOS-202234. We are pleased that you invited us to submit a revision. We have carefully attended to all comments provide by the reviewers - all of which contributed to sharpen and improve the previous version of our submission. Here we detail how we have addressed the reviewers' comments and suggestions.

We have made two major changes in the manuscript. First, there is now a new appendix in which the basic Leslie matrix theory is given, with precise definitions of the matrix elements, the dominant eigenvalue and the left and right eigenvectors. We also include the basic stochastic theory used in the model, and explain how the diffusion approximation can be defined. The main text is then changed accordingly with just a short presentation of the model referring to the appendix.

Second, much new research has been done since we wrote our manuscript, in particular estimation of transmission rates at times after infection. Originally we used a very simple model here with constant rates in an interval t_1 to t_2 , and did some sensitivity analysis varying these parameters. Now it is known that the rates follow a curve approximately proportional to a gamma distribution. We now use estimates of this from Du et al. (2020), as proposed by reviewer 2, which is a gamma with scape parameter 1.87 and mean 6.6. We keep this form in all our graphs, multiplying all transmission rates with the relevant factor to obtain different values of R_0 . Accordingly, all our figures have been updated, and we have simplified a little since sensitivity analysis on the transmission rates are no longer required. The effect of the

new transmission rates is that they change a little the relation between the dominant eigenvalue λ and R_0 , and make the first age classes having larger reproductive values.

Specific comments and suggestions from the reviewers:

Reviewer 1:

We have moved most of the Leslie matrix theory into a new Appendix A with two parts, deterministic and stochastic theory. Accordingly, much of the technicalities have been removed from the main text.

We are using the parameter R_0 , but point out that when R_E gets smaller than R_0 when a fraction of the population is immune, our results apply by simply replacing R_0 by R_E .

Reviewer 1 asks for more discussion of time to extinction when R_{eff} is close to 1. We have dealt with this by now using R_0 as large as 0.9 as reference value in the sensitivity analysis in the graphs shown in Fig.2. Accordingly, the horizontal axes now goes all the way up to 1000 days in all three graphs. Further, in the upper panel we include new lines for $R_0 = 0.95$ and 0.98 .

As proposed by the reviewer we have added a comment on new more transmissible strains that will make it harder to reduce R_0 .

Reviewer 2:

The general comment by reviewer 2 is taken care of by removing the Leslie theory to an appendix (as proposed by reviewer 1), also adding some more explanation here, as well as now using new realistic estimates of the transmission rates for the corona virus. We added a new paragraph in the beginning of the model section. We also shortened the model section using less mathematical symbols and moved the main part to an appendix. The notation is

standard in Leslie matrix theory and should be no problem.

1) More detailed explanations of the relations between the ecological and epidemiological model are given in a new section at the beginning of the modeling, explaining in particular how rates of transmission come into the model, replacing birth rates in ecology.

2) We agree that immunity of a fraction of the population no longer make R_0 the relevant parameter. However, changes in immunity is slow compared to the rather short period a person can transmit the disease. Accordingly, we point out that our analysis is valid also when $R_{eff} < R_0$ simply by reinterpreting R_0 or simply replace it by R_{eff} .

3) We have now adjusted the transmission rates so they are proportional to the gamma distribution estimated by Du et al. (2020), making our results more realistic.

4) relevant anymore. The table has been taken out.

Page 3, line 27: We have removed the NPI abbreviation which is not required.

Page 3, line 27: NPI has been removed.

Page 3, line 29: The sentence has been corrected.

Page 4, line 18: a reference to availability of vaccines has been added.

Page 4, line 27: This section has been updated.

Page 4, line 46-48: This paper addresses immigration of infections into a population, but does not attempt to evaluate the effectiveness of specific quarantine measures. An effective quarantine would be represented in our theoretical approach as a decrease in R_E

Page 4, line 52-54: The sentence has been rephrased.

Page 5, line 28: it should be clear that it is the time to extinction that follows an inverse Gaussian distribution.

Page 5, line 41: The reference has been added.

Page 6, line 16: We prefer not to introduce an additional symbol for time here. The equation explains what happens in one time step only.

The following comments are taken care of by the removal into the new Appendix A with more explanations, as well as the additional explanation in the introduction. We have now clearly pointed out that our transmission rates should be interpreted as mean values for the age group, thus in general covering effects of for example hospitalization and isolation.

Page 6, line 30-40: now this is done in Appendix A.

Page 6, line 35: Yes, the referee is right, we now use the notation for matrix transposition here.

Page 6, line 43: The explanation of the stochasticity is now better taken care of in the last part of Appendix A.

Page 6, line 51: r and V are both defined precisely.

Page 7, line 36: The concept of individual reproductive values and their variances are now explained better in Appendix A, section on the stochastic model.

Page 7, line 43: One advantage with the diffusion approximation is that it is only based on means and variances. This enables us to express over dispersion with the respect to the Poisson by a single parameter D (variance

to mean ratio). There is no need for considering specific models, such as for example the negative binomial. Its meaning is explained in Appendix A.

page 7 eq(1): The notation N_0 is the correct one for the total population size (number of transmitters). The formula expresses probabilities of extinctions before time t . We have added two horizontal lines to Fig.2 at 0.1 and 0.9 so that it is easier to see at which times the probabilities take these values.

Page 8, lines 8-22: taken care of by the new section in the introduction as well as Appendix A.

Page 8, line 24: This part has been changed. We now refer to Fig.2 (based on equation (1)) and the meaning of the parameters should be clear.

Page 8, line 29: All figures are redone, the order is changed and the legends are modified. It should now be ok.

Page 8: We actually know very little about how long the relevant immigrants have been infected. But, no symptoms may indicate that their infection on average was recent. We have in all our illustrations used that the mean reproductive values v_m of the immigrants are the mean for the first three days. However, our model is general so that v_m may be replaced by realistic values when such estimates are available.

page 9, line 28: thanks, corrected.

page 9, eq(2): we have made it clear that with immigration the process will not go extinct. The stochastic process V then reaches a stationary distribution, which is what we report by equation (2) illustrated by the lower panel of Fig.3.

page 9, line 42: no, no extinction occur under constant immigration.

Table 1: has been removed from the manuscript.

page 10, line 15: R_{eff} is now better explained in Appendix A.

page 12, line 15-16: It should be quite clear from an equation given right above that D_x is the over-dispersion in the distribution of X (variance to mean ratio).

Sincerely,

Steinar Engen and Nils Chr. Stenseth on behalf of the authors

Appendix B

Dear Royal Society Open Science Editorial Team,

Manuscript ID RSOS-202234

Thank you for the positive response to our manuscript 'The ecological dynamics of the Corona epidemics during transmission from outside sources when R_0 is successfully managed below one.'

We have made the changes proposed by the referee:

The term 'population' is replaced by 'number of infectious individuals' where that is appropriate (lines 132, 134, 143, 348, 426, 436, 438)

line 119-122: two articles are added

line 128: We have added the word 'daily' to explain that the time unit is days.

line 131: changed to 'number of infectious individuals'

line 135: This was a misunderstanding. In the population dynamics theory one uses the term 'age distribution' for the distribution of individuals among age classes. These are not absolute numbers of individuals in the classes, but the relative frequencies and hence add up to one. This is now clarified.

Line 164: This was a misprint. The small n should be a capital one, which is clearly defined. This has been changed.

Line 177 and lines 146-152: We have added a verbal explanation on how the balance between the reduction due to $R < 1$ and the increase due to immigration leads to a stochastic equilibrium which is what we describe by a stationary distribution.

line 430: missing 'value' is added.

line 595 and legend to Fig.1: We now just use the term 'growth rate r ' and remove 'on the log scale' since it obviously may be misunderstood.